# Comprehensive mutagenesis identifies the peptide repertoire of a p53 T-cell receptor mimic antibody that displays no toxicity in mice transgenic for human HLA-A*0201

Iva Trenevska, Amanda P. Anderson, Carol Bentley, Tasneem Hassanali, Sarah Wiblin, Shaun Maguire, Francesco Pezzella, Alison H. Banham*, Demin Li*

Nuffield Division of Clinical Laboratory Science, Radcliffe Department of Medicine, John Radcliffe Hospital, University of Oxford, Oxford, United Kingdom

ᴑ These authors contributed equally to this work.
* demin.li@ndcls.ox.ac.uk (DL); alison.banham@ndcls.ox.ac.uk (AHB)

## Abstract

T-cell receptor mimic (TCRm) antibodies have expanded the repertoire of antigens targetable by monoclonal antibodies, to include peptides derived from intracellular proteins that are presented by major histocompatibility complex class I (MHC-I) molecules on the cell surface. We have previously used this approach to target p53, which represents a valuable target for cancer immunotherapy because of the high frequency of its deregulation by mutation or other mechanisms. The T1-116C TCRm antibody targets the wild type $p53_{65-73}$ peptide (RMPEAAPPV) presented by HLA-A*0201 (HLA-A2) and exhibited *in vivo* efficacy against triple receptor negative breast cancer xenografts. Here we report a comprehensive mutational analysis of the p53 RMPEAAPPV peptide to assess the T1-116C epitope and its peptide specificity. Antibody binding absolutely required the N-terminal arginine residue, while amino acids in the center of the peptide contributed little to specificity. Data mining the immune epitope database with the T1-116C binding consensus and validation of peptide recognition using the T2 stabilization assay identified additional tumor antigens targeted by T1-116C, including WT1, gp100, Tyrosinase and NY-ESO-1. Most peptides recognized by T1-116C were conserved in mice and human HLA-A2 transgenic mice showed no toxicity when treated with T1-116C *in vivo*. We conclude that comprehensive validation of TCRm antibody target specificity is critical for assessing their safety profile.

## Introduction

In contrast to classical monoclonal antibodies, which recognize cell surface or secreted antigens, T-cell receptor mimic (TCRm) antibodies are able to target peptides derived from intracellular proteins [1]. These peptides are processed through the cellular proteasome-dependent and–independent mechanisms in order to be presented on the cell surface by MHC class I (MHC-I) molecules. Peptides presented by MHC-I are normally recognized by the T-cell

**Data Availability Statement:** All relevant data are within the manuscript and its Supporting Information files.

**Funding:** I.T. supported by Medical Research Council scholarship A.P.A., C.B., T.H. S.W., A.H.B, D.L. sponsored by Cancer Research UK Programme Grant A10702 and Project grant A21667 S.M. sponsored by Medical Research Council Confidence in Concept grant BRR00080 The funders had no role in study design, data collection and analysis, decision to publish, or preparation of the manuscript.

**Competing interests:** The authors have declared that no competing interests exist.

receptor (TCR) of T cells as part of the body's immune defence mechanism–a pattern of recognition that is adopted by TCRm antibodies. Some of the first TCRm antibodies against cancer antigens targeted peptides derived from MAGE-A1 and gp100 for the treatment of melanoma [2–4]. Since then, the repertoire of antibodies and targets has expanded, and includes additional examples such as the ESK1 antibody, which targets a WT1-derived peptide presented by HLA-A*0201 [5]. ESK1 has been studied extensively and the solving of its crystal structure with its target epitope has provided greater insight into its specificity and binding orientation [6]. We have previously reviewed the published TCRm antibodies and their targets [7].

We have previously reported the production of a TCRm antibody, T1-116C, which targets a wild type p53 peptide (RMPEAAPPV; p53RMP) that is presented by HLA-A*0201 [8]. Targeting the p53 tumour suppressor is appealing as p53 is widely dysregulated, p53-derived peptides are more abundantly presented by MHC-I on cancer cells and therefore p53-targeting TCRm could have broad therapeutic applications [9]. This antibody has been shown to be effective in inhibiting tumour growth both *in vitro* and *in vivo* [10]. In order to further develop T1-116C as a therapeutic, it is crucial to test its specificity and define the amino acids in the target peptide that comprise its epitope. The approach used to test specificity must take into consideration the two elements contributing to TCRm target recognition. While monoclonal antibodies typically bind epitopes from a single antigen, which may be conformation dependent, TCRm antibodies bind epitopes derived from two components: the MHC-I molecule and the peptide presented within the antigen cleft. Studies have demonstrated that their HLA allele specificity is fairly stringent, although this may advantageously include multiple HLA-A*02 subtypes [6]. TCRm antibodies tend to bind only certain key residues in the antigen peptide, thus leaving some scope for variation at the other residues within the peptide sequence [11–13], provided that the amino acid residues at the key MHC-I and antibody binding positions are preserved [6]. This could pose a safety risk if cross-reactive epitopes derived from other proteins are presented on healthy cells, thus highlighting the need to assess the extent of this potential risk. Such a hazard was exemplified in a clinical trial where a MAGE-A3-targeting affinity-enhanced TCR cross-reacted with a peptide derived from the unrelated protein titin, leading to fatal cardiac toxicity in two patients [14–16].

Here we report further specificity and safety testing of the T1-116C anti-p53 TCRm monoclonal antibody (mAb) in order to determine its suitability for cancer immunotherapy. We first performed systematic amino acid substitution studies at each position to determine requirements for peptide specificity, and identified a key residue that is crucial for antibody binding. We used the consensus derived from these studies to identify and test potentially cross-reactive peptides. As the majority of cross-reactive peptides were conserved in mice we also tested the *in vivo* safety profile of T1-116C in a human HLA-A2 transgenic mouse model.

## Materials and methods

### Cell lines and cell culture

NCI-H2087, NCI-H1395, NCI-H1930, NCI-H1299, Hs 695T, and AU565 cell lines were obtained from ATCC in 2004. SW480, Granta-519, OCI-Ly8 and KM-H2 were obtained from DSMZ in 2004 or prior to 2000. MDA-MB-231, MDA-MB-453, MDA-MB-468 and MCF7 breast cancer cell lines were obtained from Clare Hall Laboratories (CRUK) in 2004. SUDHL-6 and OCI-Ly3 were acquired from Dr Eric Davis (NIH, Bethesda) in 2000. CCRF-CEM and KARPAS-299 were acquired from Georges Delsol (Toulouse) between 2000 and 2004. Daudi and Jurkat cell lines were obtained from the William Dunn School of Pathology, University of Oxford before 2000. The 143B cell line was obtained from Judy Bastin (Oxford University) in

2015. MO1043 was acquired from Prof Riccardo Dalla-Favera at Columbia University in 2014. FL-18 was acquired from Shirou Fukahara at Kyoto University prior to 2000. Thiel was a gift from Prof Volker Diehl at the University of Cologne, Germany. MOLT-4 was obtained from the Necker Children's Hospital in Paris, France. SUDHL-1 was obtained from Steve Morris at St. Jude Children's Research Hospital prior to 2000. T2 cells lines was a gift from Prof Alain Townsend at the University of Oxford in 2009.

Cell lines were cultured in RPMI 1640 media supplemented with 10% foetal bovine serum, 1x penicillin-streptomycin [penicillin (50U/ml) and streptomycin (50µg/ml), and 1x L-glutamine (all from Gibco). Cell lines were cultured at 37˚C in a 5% $CO_2$ incubator. Breast cancer cell lines and SW480 colorectal adenocarcinoma cell line were cultured in DMEM (Gibco) supplemented as above. Cell lines were regularly tested for mycoplasma infection and experiments were conducted prior to reaching 15 passages.

## T2 stabilization assay

Synthesized peptides (Sigma-Aldrich) were dissolved in DMSO (Sigma-Aldrich) to make a stock concentration of 50mM. T2 cells were washed in serum-free RPMI 1640 and plated at $1x10^5$ cells/100µl/well, in a flat-bottom 96-well plate (Greiner Bio-One). Peptides were diluted into 100µl serum-free media and added to the cells to a final concentration of 50µM. Plates were incubated overnight for 16 hours and then harvested for flow cytometry analysis.

## Flow cytometry

T2 cells were washed with FACS wash (PBS with 2% FBS and 0.1% sodium azide), and incubated with primary antibody at 10µg/ml [isotype control mIgG1 (R&D Systems, cat#: MAB002, clone: 11711R) or mIgG2a (R&D Systems, cat#: MAB003, clone: 133304), BB7.2 (BioRad Laboratories, cat#: MCA2090, clone: BB7.2) or T1-116C (in-house)] for 30 minutes at 4˚C. Cells were pelleted and then washed before staining with secondary antibody [anti-mouse IgG-PE (Dako, cat#: R 0480), or anti-human IgG-PE (Jackson ImmunoResearch Inc., cat#: 109-116-170)] for 20 minutes at 4˚C. After a final wash the cells were fixed and analysed using a flow cytometer (FACSCaliburTM, BD Biosciences). Data was analysed using FlowJo software (Tree Star Inc.). A list of the antibodies used is available in S4 Table.

## Real-Time Quantitative Polymerase Chain Reaction (RT-qPCR)

Expression of transcripts was assessed using cDNA from normal human tissues (Clontech Human MTC™ Panels I and II, Takara Bio Europe, 636742 and 636743), and a panel of malignant cell lines. MTC™ Panel cDNAs are prepared from pools of donors. Cell line total RNA was extracted using the QIAgen RNeasy Minikit (QIAgen, 74106) according to the manufacturer's protocol. Total RNA (1µg) was reverse transcribed using Superscript® III (Life Technologies, 18080044). cDNA was diluted 1 in 5 and 4µl was used for RT-qPCR in a 20µl reaction volume. The RT-qPCRs were performed with EXPRESS qPCR Supermix, Universal (Life Technologies, 11785–200), in a 96-well plate on an MJ Research Chromo4 thermal cycler. Exon-spanning TaqMan® assays (Life Technologies) for target genes were selected to ensure detection of transcripts containing the exon encoding the cross-reactive peptide. Assay details are listed in S5 Table. Normal tissue expression was normalized to *β2M* (Beta-2-microglobulin) and *GAPDH* (glyceraldehyde-3-phophate dehydrogenase), whilst cell line expression was normalized to *TBP* (TATA Box Binding Protein), *18S RNA* and *HPRT1* (Hypoxanthine phosphoribosyltransferase 1). Expression is presented relative to positive control samples.

### Safety testing in HHD mouse model

The animal experiment was approved by the Oxford University Animal Welfare Ethical Review Body (AWERB) and governed by UK Home Office project license 30/3133. HHD mice were a kind gift from Dr Uzi Gileadi at the Human Immunology Unit, Weatherall Institute of Molecule Medicine in Oxford. All efforts were made to minimize suffering during the experiments. The female mice were between 6 and 8 weeks old at the start of the experiment. During the study, animals were monitored regularly for health and welfare signs in addition to routine husbandry care and their body weights were measured twice weekly. They were injected intra-peritoneally twice weekly, for a total of seventeen interventions, with 20mg/kg humanized T1-116CV1-mIgG2a or anti-fluorescein isotype control antibody (Absolute Antibody).

### Blood cell population analysis

Animals were sacrificed at the end of the experiment and their blood was collected. Red blood cells were lysed with lysis buffer (Sigma Aldrich, R7757) and white blood cells (WBCs) were stained with anti-CD11b-PE (BioLegend, cat#: 101207, clone: M1/70), anti-Gr1-biotin (Invitrogen, cat#: 13-5931-81, clone: RB6-8C5) and streptavidin-PerCP eFluor™710 (Invitrogen, cat#: 46-4317-82), anti-CD19-APC (Invitrogen, cat#: 17-0193-80, clone: eBio1D3) and anti-CD3-BV421 (BioLegend, cat#:100227, clone: 17A2) antibodies to quantify immune cell populations. Flow cytometry was performed using an LSRFortessa (BD Biosciences) and results were analysed using FlowJo software (Tree Star Inc.). The student's t-test was used for statistical analysis, using Prism (GraphPad software).

### Hematoxylin and eosin staining of tissues

Organs (heart, liver, kidney, lungs, spleen and colon) from each mouse were harvested and fixed in formalin (Sigma-Aldrich, HT501128) for two days, at 4˚C. Fixed organs were processed using a Tissue-Tek VIP 5Jr. (Sakura) processor and embedded in paraffin (VWR). Tissue sections (4μm thick) were cut using a microtome (Leica RM 2135) and stained with hematoxylin (Thermo Scientific, 6765009), followed by counterstaining with eosin (Sigma-Aldrich, HT110216) as per standardized protocol. Mounting medium (Vector Labs, H-5000) and cover slips (VWR) were placed on the tissues before leaving overnight to dry. Tissues were visualized using an Olympus BX51 microscope and photos acquired with an Olympus DP70 digital camera and DP controller software (Olympus). Slides were scored by a qualified pathologist who was blinded to treatment.

## Results

### Alanine and glycine scanning mutagenesis identified an initial consensus sequence for T1-116C binding

In order to test the peptide specificity and to define the epitope bound by the anti-p53 TCRm mAb T1-116C, alanine or glycine single amino acid substitutions were made across the p53RMP peptide and each of the mutant peptides were tested in T2 stabilization assays. Pulsing human T2 lymphoblast cells with peptides that can bind HLA-A2 stabilizes the HLA-A2/peptide complex on the T2 cell surface and allows peptide presentation to be studied. The peptide-pulsed T2 cells were then stained with (i) BB7.2 (anti-HLA-A2 antibody) to assess HLA-A2 presentation, and (ii) T1-116C, to assess the effect of substitution on antibody binding (Fig 1A and 1B). The positions within the p53RMP peptide where amino acid substitution by either alanine or glycine reduced T1-116C binding by >50% were used to generate the consensus sequence of RXPXXAPXV (X represents any natural amino acid) for recognition,

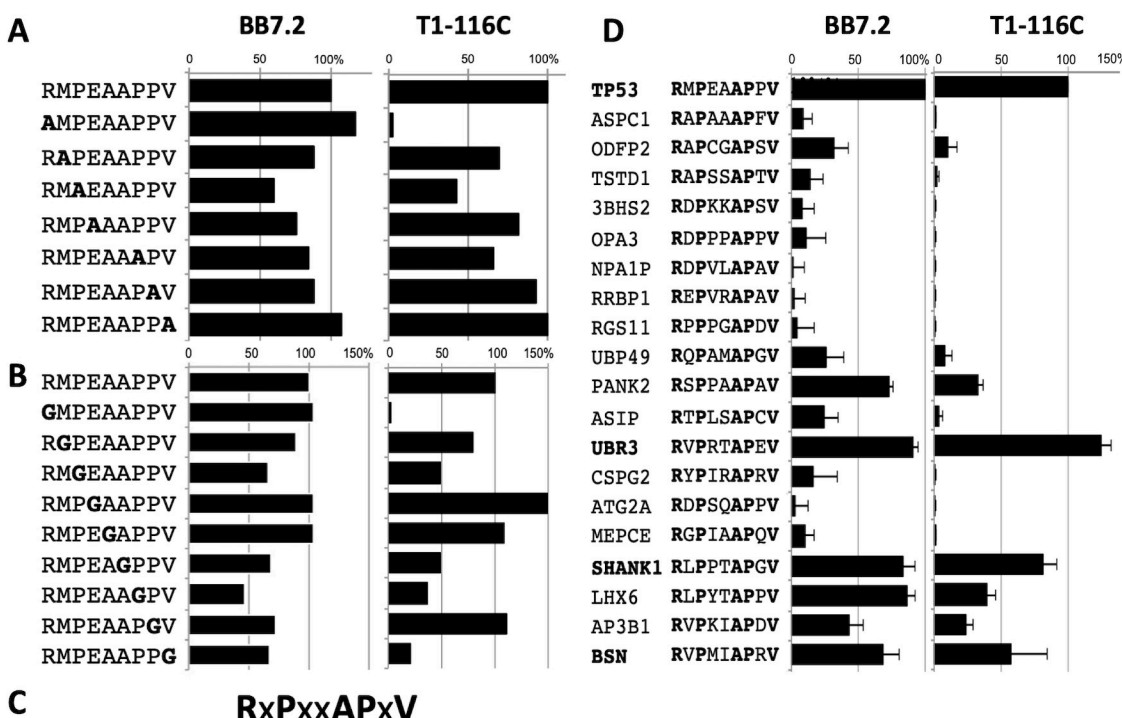

**Fig 1. Alanine and glycine scanning mutagenesis identifies an initial consensus sequence for T1-116C binding.** Individual amino acids within the p53RMP peptide were substituted with **(A)** alanine or **(B)** glycine and the mutant peptides were tested in a T2 stabilization assay for HLA-A2 mAb (BB7.2) and T1-116C binding. The bar graphs depict the mean fluorescence intensity (MFI) of antibody binding for each substituted peptide relative to the p53RMP peptide. **(C)** A T1-116C consensus recognition motif of RxPxxAPxV was deduced where X indicates a position where both substitutions retained >50% of T1-116C binding. **(D)** Human peptides matching the T1-116C binding consensus in the UniProtKB/Swiss-Prot protein database. Individual peptides were tested in T2 assays for T1-116C and BB7.2 binding. Combined data from three replicate experiments is presented. UBR3, SHANK1, and BSN peptides exhibited >50% of T1-116C binding.

where positions 1, 3, 6, 7 and 9, contributed to peptide presentation/antibody binding (Fig 1C). The effective abrogation of T1-116C binding when arginine at position 1 was substituted for either alanine or glycine, without any reduction in HLA-A2 presentation, indicated that this position is crucial for antibody binding to the p53RMP peptide.

To identify cross-reactive peptides that could potentially be recognized by the T1-116C mAb, the RXPXXAPXV consensus sequence was compared to sequences in the UniProtKB/ Swiss-Prot protein database, using the ScanProsite tool. All nineteen human peptides matching the T1-116C consensus were identified, synthesized and tested in T2 stabilization assays (Fig 1D). The majority of the peptides (13) did not effectively stabilize HLA-A2, consistent with low *in silico* predicted HLA-A*0201 binding scores from BIMAS and SYFPETHI (S1 Table). PANK2 was more effectively bound to MHC-I than predicted *in silico*, indicating the importance of experimental validation. Only three peptides, derived from the adaptor protein SH3 and multiple ankyrin repeat domains protein 1 (SHANK1), E3 ubiquitin-protein ligase (UBR3), and structural protein Bassoon (BSN), demonstrated T1-116C staining at >50% of the level observed with the wild type p53RMP peptide.

Transcript expression patterns for *SHANK1*, *UBR3*, and *BSN* in cancer cell lines and Clontech's normal human tissue panels were analysed using qPCR, detection of the exon encoding the cross-reactive peptide (Fig 2). Among cancer cell lines transcript expression for *SHANK1* was highest in NCI-H1930 (lung), 143B (osteosarcoma) and SUDHL1 (non-Hodgkin's lymphoma). *BSN* transcripts were most abundant in NCI-H1930, OCI-Ly3, SUDHL1 (all non-Hodgkin's lymphoma)

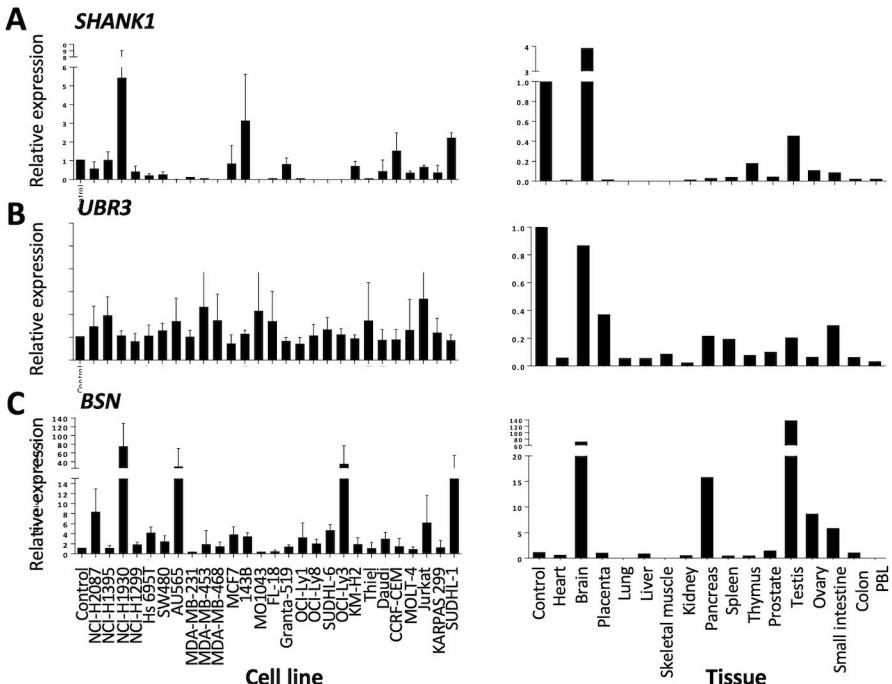

**Fig 2. Transcript expression of genes encoding T1-116C cross-reactive peptides in cancer cell lines and in normal human tissues.** Quantitative real time PCR (qPCR) analysis of transcript expression for **(A)** *SHANK1*, **(B)** *UBR3* and **(C)** *BSN* in cancer cell lines (left) and in normal human tissues (right). For the qPCR analysis of transcript expression in cancer cell lines, transcript expression was determined using a Taqman probe that corresponds to the cross-reactive peptides. Expression was normalized to TBP, 18S RNA and *HPRT1* and expressed relative to expression in control cancer cell lines (MDA-MB-453 for *UBR3* and *BSN*, or ACH-N for *SHANK1*). The coding exon for the cross-reactive peptides is exon 5 (Ex5) for *BSN*, Ex22 for *SHANK1* and *UBR3*. For the cancer cell lines, combined data from three replicate experiments is presented, whereas for the normal human tissues the MTC^TM Panels are composed of samples from at least five different tissues and the results of one representative experiment are presented. For the normal human tissues, Clontech's Human MTC^TM Panel I and II were used to analyse transcript expression using a Taqman probe corresponding to the exon encoding the cross-reactive peptide. Expression was normalized to *GAPDH* and *B2M* and expressed relative to a control cancer cell line (MDA-MB-453 for *UBR3* and *BSN*, or ACH-N for *SHANK1*).

and AU65 (breast cancer). Notably, despite being HLA-A2+, NCI-H1930 and SUDHL1 did not bind T1-116C [8]. *UBR3* transcripts were more abundant in cancer cell lines than in normal tissues. The HLA-A2[+] Thiel (p53-negative) and MDA-MB-453 (weakly p53[+]) cell lines were among the highest *UBR3* expressors, yet both lacked T1-116C binding [8].

In normal human tissues transcript expression for *SHANK1* and *BSN* is highest in the brain and testis, which are both immune privileged organs. *UBR3* transcription is more widespread in normal human tissues, with the brain showing the highest expression. It is also important to highlight that transcript expression may not correlate with protein expression or subsequent peptide processing and presentation. Importantly, the peptides from SHANK1, BSN and UBR3 were not among the experimentally determined T-cell epitopes in the Immune Epitope Database (http://www.iedb.org). Both the SHANK1 and BSN peptides are conserved in mice.

To test the validity of the RXPXXAPXV consensus sequence, we selected several well-known cancer T-cell epitopes that retained the critical arginine (R) in position 1 but mismatched other positions in the consensus sequence. T1-116C stained T2 cells presenting peptides derived from WT1, MG50, Tyrosinase, gp100 and NY-ESO-1 proteins (Fig 3). These findings suggest that T1-116C can recognize multiple cancer antigens, and that the consensus derived from *in silico* alanine and glycine scanning alone is insufficient to predict potentially cross-reactive peptides.

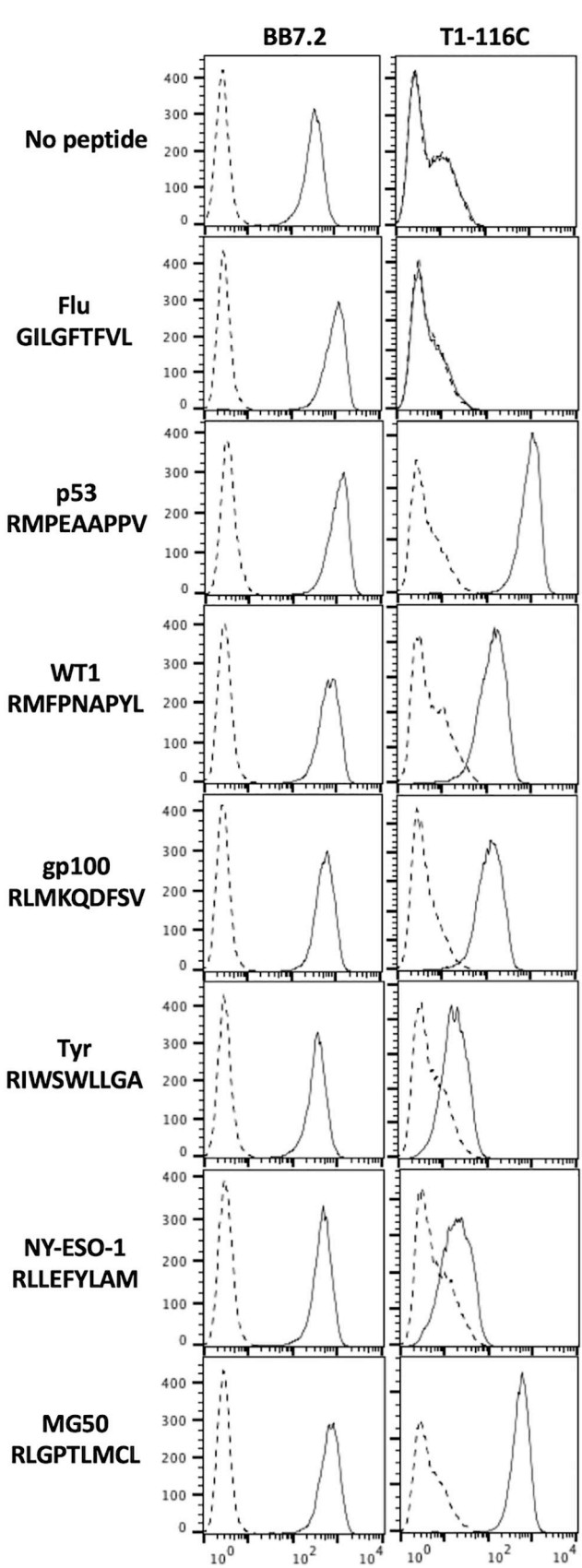

**Fig 3. Cancer-related peptides that do not comply with the initial binding consensus are recognized by T1-116C.** Peptides derived from cancer-related proteins that have an arginine at position 1 (R1), but that did not match the T1-116C binding consensus, were synthesized and tested in T2 assays. An irrelevant peptide derived from influenza A virus (GILGFTFVL) was used as a negative control. HLA-A2 expression was detected using the BB7.2 antibody. One representative experiment of three independent biological replicates is shown.

## Comprehensive amino acid substitutions of the p53RMP peptide revealed complex T1-116C antigen specificity

To assess the specificity of T1-116C in more depth, each amino acid within the p53RMP peptide was substituted individually for each of the 19 remaining essential amino acids, generating a panel of 172 peptides for testing in T2 presentation assays. Fig 4A shows that substitutions at the anchor residues (position 2 and 9), which bind MHC-I, had the greatest effect on p53RMP peptide presentation by HLA-A2 (Fig 4A) and consequently also impaired T1-116C binding as demonstrated in Fig 4B. The arginine in position 1 is absolutely indispensable for T1-116C

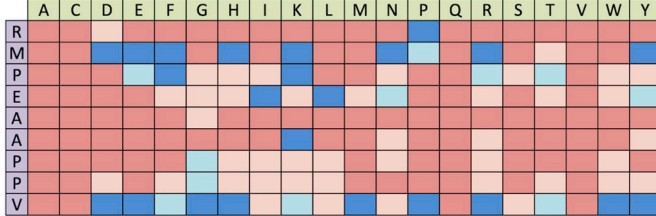

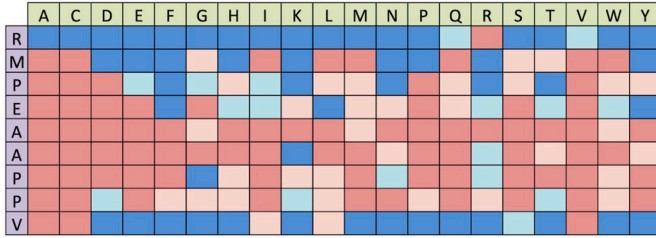

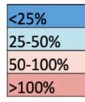

**C**

|   | A | C | D | E | F | G | H | I | K | L | M | N | P | Q | R | S | T | V | W | Y |
|---|---|---|---|---|---|---|---|---|---|---|---|---|---|---|---|---|---|---|---|---|
| 1 |   |   |   |   |   |   |   |   |   |   |   |   |   |   | R |   |   |   |   |   |
| 2 | A | C |   |   |   | G |   | I |   | L | M |   |   | Q |   | S | T | V | W |   |
| 3 | A | C | D |   |   |   | H |   |   | L | M |   | P | Q |   | S |   | V | W | Y |
| 4 | A | C | D | E |   | G |   |   | K |   | M | N | P | Q |   | S |   | V |   |   |
| 5 | A | C | D | E | F | G | H | I | K | L | M | N | P | Q | R | S | T | V | W | Y |
| 6 | A | C | D | E | F | G | H | I |   | L | M | N | P | Q |   | S | T | V | W | Y |
| 7 | A | C | D | E | F |   | H | I | K | L | M |   | P | Q |   | S | T | V | W | Y |
| 8 | A | C |   | E | F | G | H | I |   | L | M | N | P | Q | R | S |   | V | W | Y |
| 9 | A | C |   |   |   |   |   | I |   | L |   |   |   |   |   |   |   | V |   |   |

**Fig 4. Comprehensive amino acid substitution identifies a complex peptide consensus.** Each individual amino acid in the p53RMP nonamer was replaced one at a time with the essential amino acids, generating 172 peptides for testing. T2 cells were pulsed with each of the peptides and stained with (**A**) anti-HLA-A2 antibody (BB7.2) or (**B**) T1-116C. Antibody binding was categorized, relative to the binding observed with the wild type p53RMP peptide using the MFI values: enhanced (>100%), maintained (50–100%), reduced (25-50%), or severely diminished (<25%). The data represent averages of two independent biological replicates. (**C**) Summary of the amino acid substitutions at each position within p53RMP that retain T1-116C binding (≥50% the wild type peptide).

binding and comprises part of its epitope, as substitutions rarely affect MHC-I presentation. Fig 4C summarises the amino acid substitutions that maintain or enhance T1-116C binding, suggesting the antibody does not bind to the central amino acids that would commonly be recognized by a TCR. This comprehensive screen predicts that the consensus sequence is not as conservative as originally demonstrated by the alanine and glycine scanning.

A search of the Immune Epitope Database (IEDB) identified 271 experimentally validated HLA-A2-restricted peptides that matched the new T1-116C consensus displayed in Fig 4C (S2 Table). We selected 25 peptides derived from molecules that are widely expressed in normal human tissues (Table 1). The vast majority of the peptides (including all 25 selected) were also conserved in mice, thus enabling validation that species cross-reactive epitopes were recognized by T1-116C prior to testing *in vivo* antibody-dependent toxicity in a human HLA-A2 transgenic mouse model.

Before further specificity and safety testing, the T1-116C antibody was humanized by CDR grafting (Lonza, UK). Of the 16 humanized variants produced, four antibodies retained comparable affinity and peptide specificity to the parental antibody (S3 Table, S1 Fig). Two variants, T1-116CV1 and T1-116CV2, also bound similarly to cancer cell lines (S1A Fig) and T2

**Table 1. T1-116C binding of peptides retrieved from the IEDB by using the complex binding consensus.**

| Gene | Protein | Peptide | T1-116C | T1-116CV1 |
|------|---------|---------|---------|-----------|
| **TP53** | **Cellular tumor antigen p53** | **RMPEAAPPV** | +++ | +++ |
| MTF2 | Metal-response element-binding transcription factor 2 | RVPPVPPNV | +++ | +++ |
| USP9Y | Probable ubiquitin carboxyl-terminal hydrolase FAF-Y | RLWGEPVNL | +++ | +++ |
| DUSP2 | Dual specificity protein phosphatase 2 | RLDEAFDFV | +++ | +++ |
| LMNA | Prelamin-A/C | RLADALQEL | +++ | +++ |
| INPP5B | Type II inositol 1,4,5-trisphosphate 5-phosphatase | RLVGIMLLL | +++ | ++ |
| KANK2 | KN motif and ankyrin repeat domain-containing protein 2 isoform 2 | RLLDYVVNI | ++ | ++ |
| SNX4 | Sorting nexin-4 | RVADRLYGV | ++ | ++ |
| ARHGAP30 | Rho GTPase activating protein 30 | RLYDKFAEA | ++ | ++ |
| TRIP12 | E3 ubiquitin-protein ligase TRIP12 | RLLDTNPEI | ++ | ++ |
| CBX6 | Chromobox protein homolog 6 | RISDVHFSV | ++ | ++ |
| MCM7 | DNA replication licensing factor MCM7 | RLAQHITYV | ++ | + |
| G6PC2 | Glucose-6-phosphatase 2 | RLLCALTSL | + | + |
| NIPBL | Nipped-B-like protein | RLMDNSTSV | + | + |
| VRK2 | Serine/threonine-protein kinase VRK2 | RMLDVLEYI | + | + |
| KRT18 | Keratin, type I cytoskeletal 18 | RLAADDFRV | + | + |
| AKAP8L | A-kinase anchor protein 8-like isoform 2 | RMWEDPMGA | + | + |
| BRD4 | Bromodomain-containing protein 4 | RLAELQEQL | + | + |
| TRRAP | Transformation/transcription domain-associated protein | RVYERLLYV | + | + |
| UBE2L3 | Ubiquitin-conjugating enzyme E2 L3 | RLMKELEEI | + | - |
| NFIB | Nuclear factor 1 B-type | RQADKVWRL | - | - |
| MSH6 | DNA mismatch repair protein Msh6 | RLLSKIHNV | - | - |
| KDM1A | Lysine-specific histone demethylase 1A | RLLEATSYL | - | - |
| VAMP1 | Vesicle-associated membrane protein 1 | RLQQTQAQV | - | - |
| PRKD2 | Serine/threonine-protein kinase D2 | RQASLSISV | - | - |
| ACTB | Actin, cytoplasmic 1 | RMQKEITAL | - | - |

Peptides matching the complex binding consensus were identified by searching the Immune Epitope Database (IEDB). 25 peptides with broad normal tissue distributions, and sequence conservation in mice, were tested in T2 assays for binding by the murine T1-116C antibody and the humanized variant T1-116CV1. MFI comparisons indicate strong (+++), medium (++), weak (+) or no (-) binding respectively.

cells pulsed with alanine (S1B Fig) or glycine (S1C Fig) substituted peptides. T1-116CV1 was selected as the lead candidate as it had the most similar affinity and specificity profile to the parental antibody.

In T2 stabilization assays the staining intensity of T1-116C and the humanized T1-116CV1 variant were categorized as strong (+++), medium (++), weak (+) or no (-) binding. The two antibodies performed similarly and approximately a quarter of the validated epitopes were not recognized (Table 1), despite matching the consensus. Furthermore, three (WT1, NY-ESO-1 and MG50) of the five cancer-related peptides recognized by T1-116C (Fig 3) still did not match the more redundant binding consensus. Thus, even with more comprehensive *in silico* analysis, it was not possible to precisely predict the identity of individual functionally T1-116C cross-reactive epitopes.

### *In vivo* testing of T1-116CV1 in human HLA-A2 transgenic mice exhibits no toxicity

The MHC-I transgene presented in HHD mice encodes human β2-microglobulin covalently linked to the α1 and α2 domains of human HLA-A2*01, while the α3, transmembrane, and cytoplasmic domains of the molecule originate from murine H-2D$^b$ [17]. Female HHD mice were injected intraperitoneally twice weekly with T1-116CV1 or an isotype control antibody. Both antibodies were expressed in an mIgG2a isotype to maximise engagement of murine immune effector cells. Mice exhibited no signs of any adverse effects, nor significant difference in body weight between the T1-116CV1 and control groups (Fig 5A). The granulocyte and lymphocyte populations in blood taken from the T1-116CV1 and the control groups also showed no significant differences (Fig 5B). Furthermore, hematoxylin and eosin staining of tissue sections from multiple organs also revealed no histological evidence of toxicity following T1-116CV1 treatment (Fig 5C).

## Discussion

TCRm mAbs recognize a dual antigen epitope comprising both the MHC-I molecule and a bound peptide. In this study we focus on assessing the peptide specificity and potential safety of our anti-p53 TCRm antibody, T1-116C. Our key findings are that comprehensive amino acid substitution of the p53RMP peptide defines a consensus binding sequence that is more complex and matches a greater number of antigens than indicated by a consensus derived only from alanine and glycine scanning. Additional target antigens were verified experimentally and included several cancer epitopes that are actively being pursued for cancer immunotherapy, namely WT1, MG50, Tyrosinase, gp100 and NY-ESO-1. This extends the range of patients whose tumours are likely to bind T1-116C and the ability to bind multiple epitopes may also minimise the risk of generating escape variants under therapeutic selection pressure. Despite *in vitro* recognition of MHC-presented peptides derived from normal tissue antigens, T1-116C does not exhibit any toxicity in mice transgenic for human HLA-A2. We conclude that multiple methods, both experimental and computational, are required to investigate the specificity of TCRm antibodies, serving as a cautionary report highlighting the need to validate findings using complementary techniques.

Other researchers have also used crystallography to define the target epitopes of TCRm antibodies [6, 11, 18]. This method provides insight into the structure and orientation of binding, revealing which epitopes on the peptide and MHC-I molecule are recognized by the antibody. The WT1 cancer epitope RMFPNAPYL presented by HLA-A*0201 is targeted by the TCRm mAb ESK1. The crystal structure revealed that ESK1 binds an epitope on HLA-A*0201 that is conserved among various HLA-A2 subtypes, thus also expanding its application to a

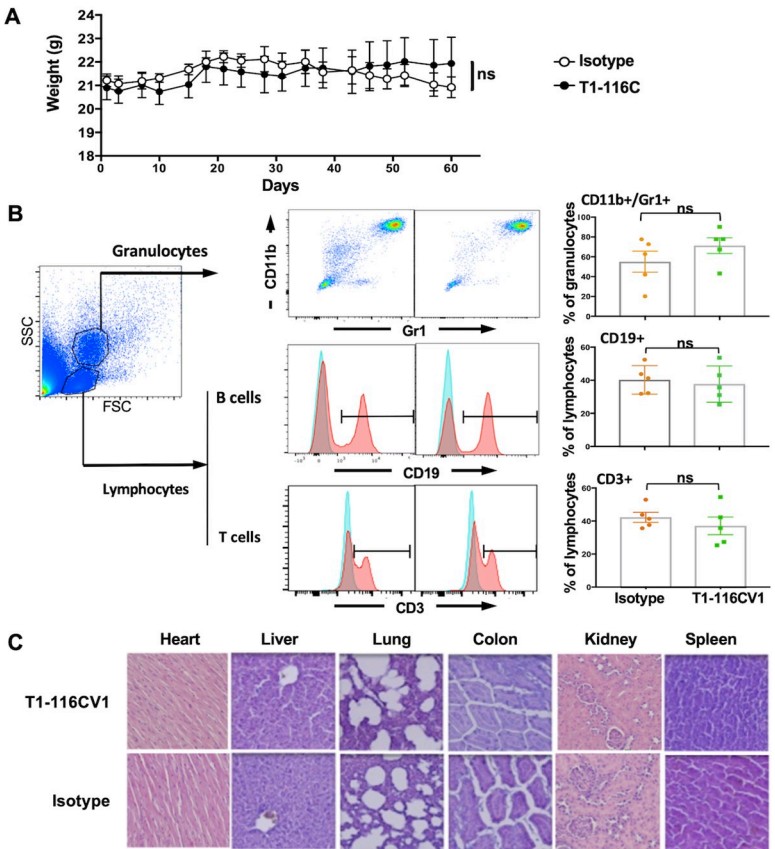

**Fig 5. T1-116C exhibits no toxicity in human HLA-A2 transgenic HHD mice.** Female HHD mice (6–8 weeks old) were injected intraperitoneally with T1-116CV1 (n = 5) or a control mAb (n = 5) at 20mg/kg, twice per week, for 8 weeks. Mice were monitored for signs of adverse effects throughout the course of treatment. (**A**) Monitoring of body weights (mean ± SEM). Mouse body weights were measured twice weekly. (**B**) Effect of antibody treatment on blood cell subsets. Antibodies against CD11b and Gr1 were used to identify the granulocytes, while antibodies against CD19 and CD3 were used to identify B and T lymphocytes respectively. The flow cytometry plots are representative of the results observed in each treatment group. No significant difference in the percentages of cells was observed between the T1-116CV1 and the control group. The bar graphs represent the mean ± SEM (n = 5). (**C**) Hematoxylin and eosin staining of organs showed no histological signs of toxicity. Representative images at 10X magnification are shown.

wider population of patients. Furthermore, ESK1 contacts three key residues at the N-terminus of the peptide: R1, F3 and P4. Structural data reveals that electrostatic forces between the contact residues and steric effects govern ESK1 binding to the peptide. As with ESK1, T1-116C also binds the N-terminus, and in particular position R1. In contrast, the crystal structure of Fab Hyb3 with its MAGE-A1 peptide presented by HLA-A*01 revealed that Hyb3 recognizes the C-terminus of the peptide and it also has a different binding orientation to ESK1 [11]. This is different to TCRs, which generally bind in a diagonal conformation that allows numerous contacts with the peptide, and in particular, with the central residues of the peptide. TCRm mAbs can bind less than half the peptide with high affinity, whereas TCRs have low affinity and their most diverse complementarity determining regions are centred over the length of the peptide, rather than at the termini. However, unlike ESK1 and Hyb3, a binding orientation similar to that observed with TCRs was demonstrated by the crystal structure of a Fab antibody that targets HLA-A*0201/NY-ESO-1$_{157-165}$. Thus, while TCRs tend to bind only in a diagonal orientation, TCRm mAbs show greater flexibility in binding orientation. Overall, the MHC portion of the pMHC complex is mostly conserved (even between subtypes), which means

that the specificity is conferred primarily by the bound peptide sequence. Therefore under-standing the specificity of TCRm mAbs for the MHC-I presented peptide is crucial for characterising their safety.

Scanning mutagenesis is an established technique used to deduce the residues within a protein that mediate protein-protein or protein-substrate interactions [19–21]. Predominantly alanine scanning mutagenesis is reported in the literature due to its favourable biological, chemical and practical properties [22–24]. Peptide scanning mutagenesis has been used to validate both the specificity of TCRs and TCRm antibodies for a target peptide [4, 25]. This is an invaluable method, however utilising only alanine and/or glycine substitution has its limitations, and as shown here, does not exhaustively identify cross-reactive epitopes.

The *in vitro* and *in vivo* techniques available for the specificity testing of TCRm antibodies are limited because of their dual epitope and requirement for human MHC class I. The ability to predict potentially cross-reactive peptides from *in silico* analyses that are naturally processed and presented, using immune epitope databases, has revolutionised the way experiments can be conducted. The volume of data that can be acquired has been increased by using high-throughput approaches such as mass spectrometry analysis of peptides eluted from MHC-I purified from normal and disease samples. This is crucial in identifying epitopes that could pose a risk to safety as it increases the chances of finding cross-reactive peptides that may otherwise be overlooked purely from *in silico* prediction of their MHC-I binding and excludes those that are not processed and presented. However, functional evaluation of hundreds of peptides is not feasible *in vitro* and it is possible that predicted off-target peptides identified in disease indications may not be processed endogenously or presented on MHC-I molecules at the cell surface in normal tissues.

The HHD mouse model has been used previously to study HLA-A2-restricted cytotoxic T-lymphocyte epitopes that are conserved in humans [26–28]. This offers a mechanism for testing potential toxicity arising from recognition of a broad range of epitopes across diverse normal tissue types. Fortunately, the majority of the predicted and experimentally validated T1-116C target peptides share sequence conservation in mice. Importantly, this approach does not limit epitopes to those that match a pre-defined consensus, which we have already shown does not fully define the T1-116C antibody specificity. However, due to the species barrier, a limitation of using HHD mice as a model is that interspecies differences in peptide sequence conservation, patterns of antigen expression, antigen processing and presentation mean that not all human epitopes can be studied [29, 30]. There is also insufficient data on transgenic protein expression level. The MAGE-A3 TCR's cross-reactivity with titin would not have been observed in this model, as the peptide is not conserved in mice [15]. Also, while we validated multiple T1-116C bound peptides that are conserved in humans and mice *in vitro*, we cannot confirm that each was presented *in vivo*. Although using the HHD mouse model has its limitations, it is an appropriate *in vivo* pre-clinical model in which to determine whether the broad range of epitopes recognized by T1-116C might cause damage to normal tissues. Crossing HHD mice to those expressing human p53 protein could enable testing of any normal tissue toxicity caused by on-target p53 binding, as the RMPEAAPPV peptide is not conserved in mice. However, the consequences of p53 targeting in normal tissues have been identified and this approach would not address other human-specific epitopes recognized by this antibody. Despite its limitations, the lack of toxicity in this *in vivo* model paves the way for further antibody testing and development.

Our aim was to define the range of peptides capable of binding the T1-116C antibody through scanning mutagenesis and *in silico* screening, in addition to assessing its safety profile *in vivo*. We discovered that comprehensive amino acid substitution, although laborious, is more effective than alanine and glycine scanning alone as a method to identify cross-reactive

epitopes. Furthermore, epitopes predicted *in silico* do not always accurately match *in vitro* findings, highlighting the need for caution and confirmation of results through different methods in order to overcome the limitations presented by each individual technique. This study emphasises the importance of testing specificity through a combination of *in silico*, *in vitro* and *in vivo* methods in order to obtain as complete a specificity profile as possible at the pre-clinical stage. Performing imaging studies in patients using a sub-therapeutic dose, or a T1-116C antibody that is unable to engage immune effector functionality, could be used to define *in vivo* antibody distribution in humans and further strengthen its safety profile.

## Supporting information

**S1 File.**
(DOCX)

**S1 Fig. Humanized T1-116C TCRm antibodies retain affinity and specificity.** (A) Binding of T1-116C humanized variants (V1-V4) to peptide-pulsed T2 cells and cancer cell lines including lymphoma OCI-Ly8, lung cancer NCI-H1395 and breast cancer MDA-MB-231. Flu peptide was used as a negative control. Blue is isotype control. Red is BB7.2 or one of the T1-116C antibodies. Peptides where (B) alanine or (C) glycine replaced the original amino acid at the indicated positions were used in T2 assays. MFI of each binding was normalized against T1-116C binding to the original p53RMP peptide. In (C) amino acids at positions 5 and 6 of p53RMP are composed of alanine therefore no alanine replacement was performed at these positions. Humanization retained binding specificity to the HLA-A2-p53RMP complex at the protein level. The humanized T1-116CV1 variant was selected for future experiments because it performs most comparably to the original murine T1-116C.
(PPTX)

**S1 Table. Predicted peptide HLA-A*0201 binding scores using BIMAS and SYFPETHI, and experimental validation using BB7.2 and T1-116C staining.**
(PPTX)

**S2 Table. IEDB peptides list.**
(XLSX)

**S3 Table. Affinity of T1-116C antibody humanization variants against HLA-A2-p53RMP complex.**
(PPTX)

**S4 Table. List of antibodies used in experimental procedures.**
(PPTX)

**S5 Table. TaqMan® assays for qPCRs.**
(PPTX)

## Author Contributions

**Conceptualization:** Alison H. Banham, Demin Li.

**Funding acquisition:** Alison H. Banham, Demin Li.

**Investigation:** Iva Trenevska, Amanda P. Anderson, Carol Bentley, Tasneem Hassanali, Sarah Wiblin, Shaun Maguire, Francesco Pezzella.

**Supervision:** Alison H. Banham, Demin Li.

**Writing – original draft:** Iva Trenevska.

**Writing – review & editing:** Alison H. Banham, Demin Li.

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
