## [Decision Letter · Decision Letter 0]

29 Mar 2021

Comprehensive mutagenesis identifies the peptide repertoire of a p53 T-cell receptor mimic antibody that displays no toxicity in mice transgenic for human HLA-A*0201

PONE-D-20-40751

Dear Dr. Li,

We’re pleased to inform you that your manuscript has been judged scientifically suitable for publication and will be formally accepted for publication once it meets all outstanding technical requirements.

Kind regards,

Scheherazade Sadegh-Nasseri

Academic Editor

PLOS ONE

Reviewers' comments:

Reviewer's Responses to Questions

**Comments to the Author**

1. Is the manuscript technically sound, and do the data support the conclusions?

Reviewer #1: Yes

2. Has the statistical analysis been performed appropriately and rigorously? 

Reviewer #1: Yes

3. Have the authors made all data underlying the findings in their manuscript fully available?

Reviewer #1: Yes

4. Is the manuscript presented in an intelligible fashion and written in standard English?

Reviewer #1: Yes

5. Review Comments to the Author

Reviewer #1: This is a well written and interesting study showing the importance of defining the fine specificity of TCR/antibodies for cancer treatment. The specificity of the antibody suggests that normal self peptides with R at P1 will also be recognized (in addition to mutated peptides in cancer patients). This raises questions about potential off-target toxicity, but the in vivo results seem important to counteract this concern.

6. PLOS authors have the option to publish the peer review history of their article (what does this mean?). If published, this will include your full peer review and any attached files.

Reviewer #1: No

---

## [Editor Report · Acceptance letter]

31 Mar 2021

PONE-D-20-40751 

Comprehensive mutagenesis identifies the peptide repertoire of a p53 T-cell receptor mimic antibody that displays no toxicity in mice transgenic for human HLA-A*0201 

Dear Dr. Li:

I'm pleased to inform you that your manuscript has been deemed suitable for publication in PLOS ONE. Congratulations! Your manuscript is now with our production department. 

Kind regards, 

on behalf of

Dr. Scheherazade Sadegh-Nasseri 

%CORR_ED_EDITOR_ROLE%

PLOS ONE